# Sequential Transcriptome Changes in the Penumbra after Ischemic Stroke

**DOI:** 10.3390/ijms20246349

**Published:** 2019-12-16

**Authors:** In-Ae Choi, Ji Hee Yun, Ji-Hye Kim, Hahn Young Kim, Dong-Hee Choi, Jongmin Lee

**Affiliations:** 1Center for Neuroscience Research, Institute of Biomedical Science and Technology, Konkuk University, Seoul 05029, Korea; adia86@naver.com (I.-A.C.); bsea78@empas.com (J.H.Y.); jh_0@naver.com (J.-H.K.); hykimmd@gmail.com (H.Y.K.); 2Department of Medical Science Konkuk University School of Medicine, Konkuk University, Seoul 05029, Korea; 3Department of Rehabilitation Medicine, Konkuk University School of Medicine, Konkuk University, Seoul 05029, Korea

**Keywords:** stroke, ischemic penumbra, transcriptome, gene profile alterations, acute stroke, chronic stroke

## Abstract

To investigate the changes in the expression of specific genes that occur during the acute-to-chronic post-stroke phase, we identified differentially expressed genes (DEGs) between naive cortical tissues and peri-infarct tissues at 1, 4, and 8 weeks after photothrombotic stroke. The profiles of DEGs were subjected to the Kyoto Encyclopedia of Genes and Genomes (KEGG) pathway and gene ontology analyses, followed by string analysis of the protein–protein interactions (PPI) of the products of these genes. We found 3771, 536, and 533 DEGs at 1, 4, and 8 weeks after stroke, respectively. A marked decrease in biological–process categories, such as brain development and memory, and a decrease in neurotransmitter synaptic and signaling pathways were observed 1 week after stroke. The PPI analysis showed the downregulation of Dlg4, Bdnf, Gria1, Rhoa, Mapk8, and glutamatergic receptors. An increase in biological–process categories, including cell population proliferation, cell adhesion, and inflammatory responses, was detected at 4 and 8 weeks post-stroke. The KEGG pathways of complement and coagulation cascades, phagosomes, antigen processing, and antigen presentation were also altered. CD44, C1, Fcgr2b, Spp1, and Cd74 occupied a prominent position in network analyses. These time-dependent changes in gene profiles reveal the unique pathophysiological characteristics of stroke and suggest new therapeutic targets for this disease.

## 1. Introduction

Stroke is the second leading global cause of death and contributes to years lived with disability worldwide [1]. Because of the improvement in the treatments available for this condition, stroke-associated mortality is declining [1]. Consequently, the majority of patients with stroke survive the acute phase and live with disability for many years [2] The only drug that is currently available to treat stroke is tissue plasminogen activator, which has a very limited time window (4.5 h) [3]; thus, rehabilitation is the only therapeutic option for patients with stroke who suffer from disability.

Neural repair arises spontaneously after stroke, continues through the semi-acute phase and slowly diminishes with time [2]. In animal models, stroke triggers the molecular program of regeneration in peri-infarct regions, which are maximally activated 1 week after stroke, followed by a plateau occurring at 3 weeks [4]. The first month after stroke is a crucial time for synaptic plasticity [4]. Consequently, most research efforts aimed at promoting recovery focus on the early phase: thus, recovery from chronic stroke is not in the spotlight. Because the long-term disability is much longer than the intensive recovery time, understanding the residual processes and finding therapeutic targets over the critical recovery period would be helpful for these patients.

Several clinical studies reported that fiber tract integrity was significantly related to motor impairment in patients with stroke [5,6]. Moreover, a significant body of evidence supports the importance of axonal reorganization in recovery after stroke [7,8,9,10,11]. The interplay between growth factors and microvascular cells prompts axonal sprouting and facilitates functional recovery [12]. The endogenous remodeling of the central nervous system is restrictive regarding the induction of complete restoration of neurological function; however, the elicitation of the endogenous restorative mechanism may represent a therapeutic avenue for this condition [12,13,14].

Despite the obvious clinical significance of post-stroke endogenous restorative changes, a detailed transcriptomics analysis of biological-process-related genes at different time points after stroke has not been performed. In the present study, we explored gene expression profiles in a peri-infarction area of a photothrombosis model via RNA-sequencing (RNA-seq) analysis. We performed functional pathway and gene ontology (GO) analyses of the differentially expressed genes (DEGs) and analyzed the protein–protein interaction (PPI) network of their products at various time points after stroke. The results allowed us to identify new therapeutic candidates that are crucial for enhancing neurorestoration and extending the therapeutic time window after stroke.

## 2. Results

### 2.1. Measurement of Infarction Volume and Neurological Deficit at Various Time Points after Stroke

The extent of the lesions in all stroke groups was significantly different from that of the control group (*p* < 0.001; one-way ANOVA). There was no statistical difference in stroke size among the stroke groups (1 week, 76.91 ± 2.29 mm^3^; 4 weeks, 84.27 ± 7.46 mm^3^; 8 weeks, 86.92 ± 7.54 mm^3^; one-way ANOVA), indicating the absence of temporal changes in lesion size (Figure 1A,B). The motor outcome was evaluated in rats using the modified neurological severity score (mNSS). mNSS scores ranged from 0 up to 14. High scores indicate that the rats had more neurological deficits from stroke. At 1, 4, and 8 weeks after stroke, the mNSS scores were 11.0 ± 0.0, 10.7 ± 0.52, and 11.0 ± 0.0, respectively, with no difference between 1, 4, and 8 weeks after stroke (Figure 1C).

### 2.2. RNA-Seq Analysis of Gene Expression Profiles in Photothrombotic Ischemic Stroke

We analyzed the transcriptome of the contralateral and ipsilateral motor cortex at different phases in photothrombotic ischemic rats using RNA-seq.

A profile of mRNA expression was displayed in the heat map (Figure 2A). Color keys indicate the relative abundance of genes in Fragments Per Kilobase of exon per Million fragments mapped (FPKM) of the samples. Messenger RNA expression of the contralateral and ipsilateral injured cerebral cortex was classified into three groups at 1, 4, and 8 weeks after stroke compared with the control group (matched for age). According to heat map analysis, the experimental groups were as follows: (1) 1-, 4-, and 8-week control cortex and contralateral cortex 1, 4 and 8 weeks after stroke; (2) ipsilateral cortex 4 and 8 weeks after stroke; and (3) ipsilateral cortex 1 week after stroke. Gene expression patterns in the contralateral cortex of 1, 4, 8 weeks after stroke exhibited were similar to the control group. Therefore, the contralateral cortices were excluded from subsequent analyses, and the changes in mRNA expression on the ipsilateral injured side were mainly analyzed.

The number of upregulated and downregulated genes (1.5-fold) was calculated and presented in Figure 2B. The DEGs in the infarction area compared with age-matched controls were as follows: 1 week post-stroke, 1950 genes were upregulated and 1821 genes were downregulated; 4 weeks post-stroke, 113 genes were upregulated and 423 genes were downregulated; and 8 weeks post-stroke, 100 genes were upregulated and 433 genes were downregulated (Figure 2B). The top five significantly upregulated or downregulated genes in the ipsilateral peri-infarct cortex of the 1-, 4-, and 8-weeks post-stroke animals are highlighted in Figure 2C,D,H and Table 1.

### 2.3. Functional Enrichment Analysis of DEGS-Temporal Changes in the GO Analysis

GO term enrichment analysis was performed to identify key biological factors that play a functionally important role in DEGs after stroke.

At 1 week post-stroke, DEGs on the most predominant responses of biological processes were associated with cellular response, apoptotic process, synaptic-vesicle exocytosis, and postsynaptic membrane potential (Figure 3A). In contrast, at 4 and 8 weeks post-stroke, we found that DEGs in biological processes predominantly correlated with the regulation of cell population proliferation, wound healing, response to estradiol, aging, and response to drugs (Figure 3B) and the negative regulation of multicellular organismal process, response to estradiol, cellular response to lipids, positive regulation of cell migration, and cell adhesion (Figure 3C), respectively.

### 2.4. Analysis of the Functional Genes in the Ontology of the Biological Processes

A PPI network analysis was performed to identify functional genes involved in the top 10 biological processes. Among the whole group of genes linked to each category (Table 2 and Appendix A), the genes corresponding to the top 10 nodes are shown. At 1 week post-stroke, DEG changes in the brain-development category revealed that *Tp53*, *Casp3*, *Rhoa*, and *Ccr5* were upregulated, and *Gnai1*, *Syt1*, *Grin2a*, and *Gnao* were downregulated. Most genes in the memory (*Bdnf*, *Gria1*, *Grin2b* and *2a*, and *Slc17a7*), synaptic-vesicle exocytosis (*Syt1*, *Stxbp1*, *Stx1a*, and *Rab3a*), postsynaptic membrane potential (*Gria1*, *Grin2a*, *Grm5*, and *Gabra1*), and cation channel activity (*Dlg4*, *Cacna1a*, *Cacng8*, *Cacng7*, and *Shank1*) categories were downregulated. Genes related to the cellular response to drugs, apoptotic process, aging, and response to state or activity of a cell were upregulated. At 4 weeks post-stroke, functionally connected genes pertaining to cell population proliferation (*Igf1*, *Ptprc*, *Timp1*, *Cxcl12*, and *Mmp2*), wound healing (*Timp1*, *Igf1*, *Sparc*, *Tgfb1*, and *Col1a1*), response to estradiol (*Igf1*, *Col1a1*, *Igf2*, *Mmp2*, and *Tgfb1*), response to organisms (*Igf1*, *Spp1*, *Col1a1*, *Fos*, and *Tgfbr2*), aging (*Igf1*, *Tgfb1*, *Timp1*, *Mmp2*, and *Fos*), response to cytokines (*Cxcl12*, *Cd44*, *Mmp2*, *Spp1*, and *Col1a1*), multicellular organismal processing (*Igf1*, *spp1*, *Mmp2*, *Col1a1*, and *Sparc*), and nervous system development (*Cd44*, *Igf1*, *Mmp2*, *Spp1*, and *Cxcl12*) were upregulated. *Igf1*, *Mmp2*, and *Col1a1* exhibited the highest degree of overlapping. These genes may be associated with increased collagen synthesis and the remodeling of the extracellular matrix (ECM) by the insulin-like growth factor (IGF-1) [15]. At 8 weeks post-stroke, genes pertaining to the negative regulation of multicellular organismal processing (*Cd44*, *Timp1*, *Ccl2*, *Tgfb1*, and *Spp1*), response to estradiol (*Ccl2*, *Col1a1*, *Igf2*, *Tgfb1*, and *Vim*), cellular response to lipids (*Ccl2*, *Icam1*, *Cd40*, *Cxcl13*, and *Tgfb1*), cell migration (*Cxcl12*, *Ccl2*, *Icam1*, *Cxcl13*, and *Ccl19*), cell adhesion (*Cd44*, *Spp1*, *Cxcl12*, *Icam1*, and *Co3a1*), ECM organization (*Col1a1*, *Col3a1*, *Lum*, *Postn*, and *Fbln1*), aging (*Ccl2*, *Tgfb1*, *Timp1*, *Icam1*, and *Clu*), cell population proliferation (*Cd44*, *Lgals3*, *Ccl2*, *Tgfb1*, and *Tgfbr2*), immune response (*Ccl2*, *Icam1*, *Timp1*, *Cd44*, and *Spp1*), and wound healing (*Timp1*, *Col1a1*, *Tgfb1*, *Dcn*, and *Cd44*) were downregulated. The *Ccl2*, *Tgfb1*, *Cd44*, and *Icam1* genes overlapped as functional genes in biological process ontology. These genes may be involved in inflammatory response and tissue remodeling [16] at a late phase after a stroke.

### 2.5. Functional Enrichment Analysis of DEGS-Temporal Changes in the KEGG Analysis

To analyze the pathways defined by these genes, we used the KEGG pathway database of DAVID [17] to classify the whole group of DEGs discovered here based on gene function information. These results were statistically significant (*p* < 0.05) and are depicted in Figure 4.

At 1 week post-stroke, the enriched KEGG pathways highlighted the pathways including glutamatergic synapse, retrograde endocannabinoid signaling, GABAergic synapse, circadian entrainment, and dopaminergic synapse compared to the sham control. Theses pathways are involved in biological functions such as synaptic transmission and function and cell adhesion (Figure 4). At 4 weeks post-stroke, pathways related to phagosomes, complement and coagulation cascades, antigen processing and presentation, osteoclast differentiation, and ECM–receptor interaction were enriched compared with the sham control. At 8 weeks post-stroke, the enriched KEGG pathways associated with antigen processing and presentation, phagosomes, complement and coagulation cascades, ECM–receptor interaction, and cell adhesion showed significant changes in comparison with the sham control.

Enriched KEGG pathways, such as those pertaining to phagosomes, complement and coagulation cascades, and ECM–receptor interaction (which are related to inflammatory responses after stroke), were altered at 4 and 8 weeks after stroke compared with that observed 1 week post-stroke (Figure 4). 

### 2.6. PPI Network Analysis

To identify molecules that are functionally related to both a GO term and a KEGG pathway, DEGs were analyzed via a PPI network analysis aimed at identifying genes with a high correlation. The top 10 and total gene products in the PPI network of the DEGs discovered here are listed in Table 3 and Appendix A, respectively. At 1 week post-stroke, the top 10 gene products were *Dlg4* (degree = 93), *Bdnf* (degree = 84), *Gria1* (degree = 81), *Rhoa* (degree = 76), *Mapk8* (degree = 76), *Gng3* (degree = 74), *Grin2a* (degree = 73), *Gnb5* (degree = 72), *Ptprc* (degree = 72), and *Tp53* (degree = 70). The disks large homolog 4 (*DLG4*) gene, which showed the highest change, encodes the post-synaptic density protein 95 (PSD-95; also known as synaptic-related protein 90 (SAP-90)). A neurotrophic factor, BDNF, the glutamate ionotropic receptor AMPA type subunit1 (Gria1), and the glutamate NMDA receptor subunit (Grin2a) were downregulated at 1 week post-stroke. These changes represent the downregulation of the functions of neurotrophic factors and glutamatergic receptors in memory and synaptic transmission after stroke. The upregulated Rhoa and Ptprc proteins may be involved in cell growth and cell-cycle progression in brain development after stroke. In turn, the upregulated Tp53 protein may be involved in the cellular regulation of the apoptosis and aging processes at the acute phase after stroke.

At 4 weeks post-stroke, the top 10 gene products in the PPI network were *Cd44* (degree = 17), *C1qb* (degree = 15), *Fcgr2b* (degree = 14), *Spp1* (degree = 12), *Cd74* (degree = 12), *C4a* (degree = 12), *C1qa* (degree = 12), *C3* (degree = 10), *Cd14* (degree = 10), and *Itgb2* (degree = 10). At 8 weeks post-stroke, the top 10 genes were *Cd44* (degree = 14), *Fcgr2b* (degree = 13), *C1qb* (degree = 13), *Cd74* (degree = 12), *C1qc* (degree = 11), *Cd14* (degree = 10), *C4a* (degree = 10), *Spp1* (degree = 10), *RT1-A2* (degree = 9), and *Itgb2* (degree = 9). The representative changes in the molecules identified in the PPI analysis exhibited similar patterns at 4 and 8 weeks. We confirmed that quantitative PCR was used to identify changes in expression of these major DEGs (Figure 5). On the whole, quantitative RT-PCR data and RNA-seq results showed high concordance coefficients, indicating that the RNA-seq results were very reliable.

## 3. Discussion

Although several studies have shown that ischemic stroke may affect the profiling of gene expression in the brain [18,19,20,21,22], a comparative analysis of the underlying gene profiles in the acute, subacute, and chronic phases of stroke has not been performed. The primary purpose of this study was to understand the differences in gene expression that occur over time after stroke. Based on previous studies demonstrating time-specific differences after injury in the rat photothrombosis model [23], we distinguished systematically the brain after stroke at the acute (1 week post-injury), subacute (4 weeks post-injury), and chronic (8 weeks post-injury) phases. The advantage of the photothrombosis model is the ability to generate an injury with a similar size within distinct functional areas of the cortex; thus, this model is suitable for assessing the overall pattern of genetic alterations that occur after stroke [24,25]. Spontaneous recovery processes together with behavioral training and experience promote recovery after stroke [26]. Therefore, functional recovery may be stimulated through the implementation of post-stroke rehabilitation and the modification of global genetic patterns. For this reason, the characteristics of the photothrombotic model are suitable for identifying and comparing changes in endogenous genes after stroke, for functional recovery.

In the present study, we used DEG, GO, associated KEGG pathway, and PPI analyses to elucidate the progression of biological processes in the peri-infarct area at various time points after stroke.

One week after stroke, 1821 and 1950 genes were significantly upregulated and downregulated, respectively. The top five increased genes were *Tmsb 10*, *Ftl1*, *Ctsb*, *Ctsd*, and *B2m*, whereas the top five reduced genes were *Mbp*, *S100b*, *Camk2n1*, *Aldoc*, and *Ndrg2*. Changes in genes involved in brain development, memory, response to drugs, apoptosis, and synaptic processes have emerged. Major synaptic transmitter and signaling pathways were dramatically changed at 1 week post-stroke. Variations in glutamatergic, GABAergic, dopaminergic, and cholinergic synapses, as well as in oxytocin, cAMP, and chemokine signaling pathways, were involved in the processes that occurred at this time point. Moreover, a PPI analysis in the GO category performed 1 week after stroke showed the upregulation of apoptosis-related molecules, such as *Tp53* and *Casp3*, while synaptic activity-related proteins were mainly decreased. At 4 and 8 weeks after stroke, the GO analysis and KEGG pathways exhibited very similar results. The PPI analysis in the GO category indicated that genes primarily related to cell population proliferation, aging, and inflammatory responses were expressed positively or negatively. The *Lgals3*, *Igf1*, *Col1a1*, and *Spp1* genes were highly upregulated.

The *Lgals3* gene, which encodes galectin-3, plays an important role in acute and chronic inflammation [27]. The insulin-like growth factor 1 (IGF-1) gene has insulin-like functions and encodes proteins involved in regulating the development and maintenance of the nervous system [28,29].

An experimental study of ischemic stroke found that galectin-3 is overexpressed in activated microglial cells and is involved in the inflammatory response in stroke [27,30]. In adult stroke models, galectin-3-positive microglial cells also produce neurotrophic factors, such as IGF-1, which also protect against damage after stroke [27,28]. Thus, the implications of increased expression of *Lgals3* at 4 or 8 weeks after stroke may include two aspects: the chronic inflammatory response and the protective effect induced by IGF-1 production after stroke or brain injury.

Ultimately, we tried to analyze protein-linked genes that were correlated with GO and KEGG pathways to identify genetic factors that are involved in the functionally significant changes in interprotein network responses that occur after a stroke. The results showed different patterns of gene expression between the time points of 1 week after stroke and 4 and 8 weeks after stroke. We showed that the expression of *Dlg4*, *Gria1*, *Grin2a*, *Gng3*, *Gnb5*, *MapK8*, and *Bdnf* (encoding PSD-95, GluA1, GluNMDA2A, G-protein subunit gamma 3, G-protein subunit beta 5, JNK, and BDNF, respectively) was strongly reduced 1 week after stroke. These findings suggest that glutamatergic receptors, membrane-associated molecules, and neurotrophic factors are involved in the acute phase of stroke. In contrast, the expression of *Rhoa*, *Ptprcm*, and *Tp53* (encoding Arha, CD45, and p53, respectively), which are involved in the regulation of lymphocytes and apoptotic cell death, was increased 1 week after stroke. In contrast, at 4 and 8 weeks after stroke, the expression of *Cd44*, *Spp1*, and *Cd74* (encoding CD44, Spp1, and CD74, respectively) was increased. Of note, spp1, which is an adhesive glycoprotein, was tremendously upregulated and this tendency continued even in the late phase. Spp1 is a multifunctional acidic phosphoprotein [31] with a controversial function. It interacts with various integrins involved in a variety of neuroprotective processes [31], and a recent study reported that macrophages produce spp1 and induce astrocyte process extension toward the infarct area, which might encourage the repair of the ischemic neurovascular unit [32]. Moreover, when bound to CD44, spp1 induced the activation of microglia and macrophages, which are associated with cytokine production and inflammation in the early phase after ischemic injury [16].

Our research identified alterations in the gene expression profiles of biological processes at various time points after stroke. Stroke transcriptomics research has focused mainly on early and subacute time points [18,19,20,21,22]. A critical period for recovery exists [26]; however, most patients with stroke endure their disability for a long period [1]. Our findings suggest that functional gene expression patterns vary greatly over time after stroke and may be applied as new therapeutic targets through specific gene regulation at these time points. These findings will help to extend the therapeutic time window for stroke.

## 4. Materials and Methods

### 4.1. Animals

Animal experimental procedures were approved by the Animal Experiment Review Board of Institutional Animal Care and Use Committee (IACUC) of Konkuk University (Permit Number: KU17042, 03 March 2017). All experiments, including treatment, anesthesia and euthanasia, were conducted in accordance with ARRIVE guidelines. They also followed the criteria for Stroke Therapy Academic Industry Roundtable (STAIR) for preclinical stroke investigations [33]. Adult male Wistar rats, 8 weeks old (283.88 ± 2.06 g), were purchased from the Orient Bio Incorporation (Seongnam, Korea). All rats were housed in a temperature-controlled room (23 ± 0.5 °C) with 12 h:12 h light/dark cycle. Food and water were provided ad libitum. Male rats were used because of the high incidence of stroke in men and to rule out the protective effects of female hormones [34,35,36,37]. Based on previous infarct volume data using an identical surgical design (photothrombotic ischemic surgery), a sample size calculation (power = 0.8; α = 0.05) estimated 6 animals per group using G-power version 3.1.9.4. For RNA sequencing, 5 animals were randomly selected to each groups and 6 animals/group were using for infarction measurement and modified neurological severity score (mNSS) (Figure 1A). Animals and samples were number-coded and investigators were always blinded to the treatment groups until the end of the data analysis.

### 4.2. Photothrombotic Stroke Model

Sensorimotor infarcts were produced by photothrombosis surgery. Animals were anesthetized with ketamine (50 mg/kg) and xylazine (5 mg/kg) mixed cocktail through intraperitoneal (i.p.) injection then their head were fixed at a stereotaxic frame (Stoelting Co., Wood Dale, IL, USA). The skull was exposed, and the fiber optic bundle of a KL1500 LC cold light source (Carl Zeiss, Jena, Germany) with a 4 mm diameter was placed on the skull 4.0 mm lateral to bregma over the right sensorimotor cortex [38]. The Rose Bengal (Sigma, Saint Louis, USA) was injected via intraperitoneal injection for 5 min (10 mg/kg) prior to light being turned on. Then, the light was switched on and persisted for 20 min. Rats were returned to cages after waking from anesthesia. Brain sampling was performed on 1, 4, 8 weeks post-stroke. The infarction core and penumbra of the photothrombotic injury were dissected and the corresponding areas in contralateral side of injured rat and from both sides of the rats were also collected.

### 4.3. Determination of the Infarction Volume

After anesthesia of animal, they were perfused with 0.9% normal saline and 4% paraformaldehyde solution in 0.1 M phosphate buffer (pH7.4). The brains were removed, post-fixed in the 4% paraformaldehyde solution for 24 h then transferred in a 30% sucrose solution in PBS for 48 h at 4 °C. The brains were cut in a coronal section (40 μm) using a cryostat at −20 °C, then stored in the −80 °C with anti-freezing solution. A total of 9 sections were collected (every 20th slice) on slides, and stained with 0.5% Cresyl violet solution. The intact areas of ipsilateral and contralateral hemispheres were measured using ImageJ software, then the volume of intact hemisphere was calculated (intact area: × 0.04 × 20) and summed among the slices. The total infarction volume was determined: the volume of contralateral hemisphere in comparison to the volume of intact area in the ipsilateral hemisphere [23,38].

### 4.4. Modified Neurological Severity Score

Animals were examined with modified neurological severity score (mNSS) at 1 week, 4 and 8 weeks after stroke. This evaluation was performed by blinded tests. mNSS consists of a motor, sensory, reflex and balance test. Scores ranged from 0 to maximum of 14 [23]. When the rat was unable to perform the task due to neurological dysfunction, it was given one point for each task. The higher score indicates severe neurological dysfunction. Motor tests have two categories. One of the categories involved raising the rat by holding its tail and observing their forelimb and hindlimb. When the rat flexed their injured forelimb and hindlimb to their body, it was given one point for each item. Another motor test evaluated gait function. The rat was placed on the floor and its walking performance was observed. When the rat walked in a normal way, the score was zero; if not, there were several points allocated depending on the severity. This test also evaluated sensory function. The examiner held and moved the rat to the corner of desk and let it touch the corner using its forepaws. If the rat counter-pushed the corner, it had zero points for the proprioceptive sensory test; if not, and the rat had malfunction in its proprioception, it was allocated one point. Balance tests were performed on a wood beam. While the rat was walking on the beam, the time was recorded until the rat fell or a maximum of 1 min had passed, and the experimenter observed gait function. Crossing the beam with steady posture was zero, but grasping or hugging the beam, or spinning and falling off the beam earned points. Reflex tests were about pinna, corneal and startle reflex. The rat was gently held and had its auricle touched; if the rat did not show any response, one point was given. The corneal reflex was examined using soft cotton. If the rat did not blink its eyes when the experimenter lightly touched the cornea with the cotton, it was given one point. The experimenter then suddenly clapped and observed the response of the rat. When the rat was startled by a sudden noise, the score was zero; if not, the score was one [23].

### 4.5. RNA Extraction and RNA-Sequencing Analysis

After the collected tissues were homogenized, total RNA was extracted using Trizol reagent (Invitrogen, Carlsbad, CA, USA). Total RNA samples were converted into cDNA libraries using the TruSeq Stranded mRNA Sample Prep Kit (Illumina, San Diego, CA, USA). Starting with 1000 ng of total RNA, poly-adenylated RNA (primarily mRNA) was selected and purified using oligo-dT-conjugated magnetic beads. This mRNA was physically fragmented and converted into single-stranded cDNA using reverse transcriptase and random hexamer primers, with the addition of Actinomycin D to suppress DNA-dependent synthesis of the second strand. Double-stranded cDNA was created by removing the RNA template and synthesizing the second strand in the presence of dUTP (deoxyribouridine triphosphate) in place of dTTP (deoxythymidine triphosphate). A single A base was added to the 3′ end to facilitate ligation of sequencing adapters, which contain a single T-base overhang. Adapter-ligated cDNA was amplified by a polymerase chain reaction to increase the amount of the sequence-ready library. During this amplification, the polymerase stalls when it encounters a U base, rendering the second strand a poor template. Accordingly, amplified material used the first strand as a template, thereby preserving the strand information. Final cDNA libraries were analyzed for size distribution and using an Agilent Bioanalyzer (DNA 1000 kit; Agilent, Santa Clara, CA, USA), quantitated by qPCR (Kapa Library Quant Kit; Kapa Biosystems, Wilmington, MA, USA), then normalized to 2 nmol/L in preparation for sequencing [39,40]. Indexed libraries were then sequenced using the HiSeq4000 platform (Illumina, San Diego, USA) by the Macrogen Incorporated. We preprocessed the raw reads from the sequencer to remove low quality and adapter sequence before analysis and aligned the processed reads to the *Rattus norvegicus* (rn6) using HISAT v2.0.5 [41]. The reference genome sequence of *Rattus norvegicus* (rn6) and annotation data were downloaded from the UCSC table browser (http://genome.uscs.edu). Transcript assembly and abundance estimation was obtained by using StringTie [42,43]. After alignment, StringTie v1.3.3b was used to assemble aligned reads into transcripts and to estimate their abundance. The relative abundances of gene were measured in FPKM (Fragments Per Kilobase of exon per Million fragments mapped) using StringTie. We performed the statistical analysis to find differentially expressed genes using the estimates of abundances for each gene in samples. Genes with one more than zeroed FPKM values in the samples were excluded. To facilitate log2 transformation, 1 was added to each FPKM value of filtered genes. Filtered data were log2-transformed and subjected to quantile normalization. Statistical significance of the differential expression data was determined using an independent *t*-test and fold change in which the null hypothesis was that no difference exists among the groups. False discovery rate (FDR) was controlled by adjusting *p* value using Benjamini–Hochberg algorithm [44,45,46].

### 4.6. Pathway Analysis

DEG lists at each time point were annotated to Kyoto Encyclopedia of Genes and Genomes (KEGG) pathway to identify significant pathway. Pathways with a FDR-adjusted *p*-value ≤ 0.05 were filtered. The Bonferroni correction was applied to adjust the *p*-value. Pathways were considered significantly enriched if the *p*-value was less than 0.05. ClueGO plugin of Cytoscape was used to generate graphical network of pathways [17].

### 4.7. Gene Ontology (GO) Analysis

All DEGs at each time point were analyzed by the Gene Ontology database (http://www.geneontology.org/). Biological process with a FDR threshold of ≤0.05 were filtered and biological process-related categories were selected and grouped by hierarchy.

### 4.8. Identification of Key Genes from Protein–Protein Interaction (PPI) Analysis

To determine relevant genes of each time points, we selected DEGs in the significantly enriched in GO terms and KEGG pathways (*p* < 0.05). Those selected genes were analyzed by Search Tool for the Retrieval of Interacting Genes (STRING), (http://string-db.org) revealing protein–protein interaction and the criteria for minimum required interaction score was medium confidence (>0.400) and FDR < 0.05. The number of interaction was counted and the genes acquired top scored interactions were regarded as potential key genes.

### 4.9. Total RNA Extraction and RT-PCR Analysis

Total RNA was extracted from cortical tissues using Trizol reagent (Invitrogen, Carlsbad, CA, USA). Reverse transcription was performed for 1 h at 42 °C with1 µg of total RNA using 20 unit/µL of AMV reverse transcriptase (Roche Applied Science, Indianapolis, IN, USA), and oligo-p(dT)15 as a primer. The samples were then heated at 99 °C for 5 min to terminate the reaction. The cDNA obtained from 1 µg total RNA was used as a template for PCR amplification. Oligonucleotide primers were designed based on Genebank entries (Table 4). PCR mixes contained 10 µL of 2 × PCR buffer, 1.25 mM of each dNTP, 10 pmol of each forward and reverse primer, and 2.5 units of Taq polymerase in the final volume of 20 µL. Amplification was performed in 35 cycles at 60 °C, 30 s; 72 °C, 1 min; 94 °C, 30 s. After the last cycle, all samples were incubated for an additional 10 min at 72 °C for final extension step. PCR fragments were analyzed on 1.2% agarose gel in 0.5 × TAE containing ethidium bromide. Amplification of GAPDH, a relatively invariant internal reference RNA, was performed in parallel, and cDNA amounts were normalized against GAPDH mRNA levels. The primer set specifically recognized only the gene of interest as indicated by amplification of a single band of expected size [47,48].

### 4.10. Statistical Analysis

The graphs represented mean ± SEM in infarction volume measurement, mNSS evaluation and RT-PCR analysis. A one-way ANOVA test with Tukey′s multiple comparisons test (all were shown in the figure legend) were used to determine the significance difference between different groups. The data was analyzed using GraphPad Prism version 7 (GraphPad software, San Diego, CA, USA). In the KEGG pathway, the *p*-value corrected with Bonferroni’s step-down. GO analysis used Fisher’s exact test with a FDR multiple test correction.

## Figures and Tables

**Figure 1 ijms-20-06349-f001:**
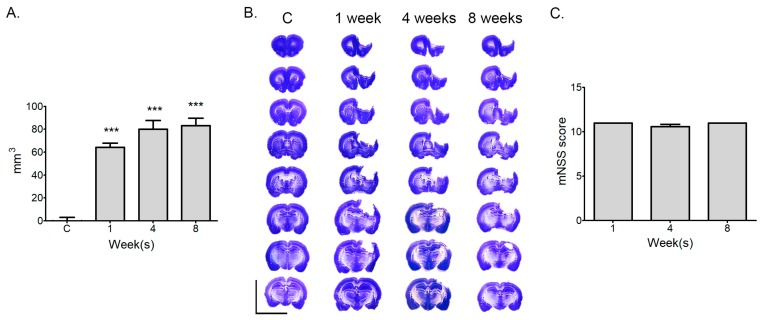
Evaluations of infarct volume and motor function from acute to chronic stages after a photothrombic ischemic stroke: (**A**) Quantification of infarction size did not differ among time points after a photothrombic ischemic stroke. Results are presented as the mean ± SEM, *n* = 6; *** *p* < 0.001 compared to control, one-way ANOVA. (**B**) Representative photomicrography of Nissl-stained sections at several time points after a stroke, Scale bars = 10 mm. (**C**) Modified neurological severity scores (mNSS) were not differ at different time points. Results are presented as the mean ± SEM, *n* = 6.

**Figure 2 ijms-20-06349-f002:**
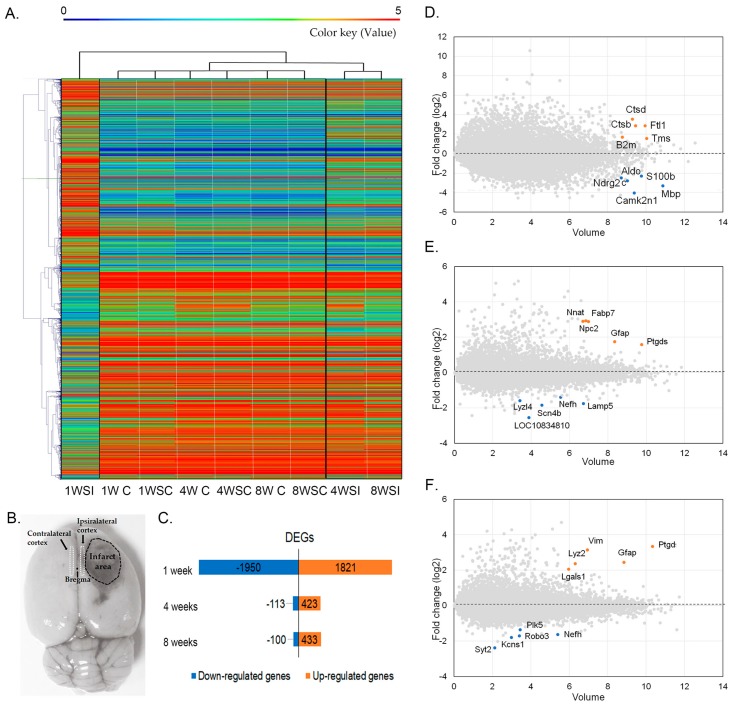
Heat map and differentially expressed genes (DEGs) of RNA-seq analysis. (**A**) RNAseq transcript heat map of Log2 (FPKM) of controls and stroke ipsilateral side and contralateral side at 1 week, 4 weeks and 8 weeks post-stroke; 1WSI: ipsilateral cortex at 1 week post-stroke; 1W C: cortex at 1 week sham control; 1WSC: contralateral cortex at 1 week post-stroke; 4W C: cortex at 4 weeks sham control; 4WSC: contralateral cortex at 4 weeks post-stroke; 8W C: cortex at 8 weeks sham control; 8WSC: contralateral cortex at 8 weeks post-stroke; 4WSI: ipsilateral cortex at 4 weeks post-stroke; 8WSI: ipsilateral cortex at 4 weeks post-stroke; FPKM: Fragments Per Kilobase of exon per Million fragments mapped. (**B**) Sampling area of contralateral and ipsilateral cortex after stroke (coordination: AP 2 mm to –1mm; ML 0 to 2 mm; DV 2.5 mm, M1 and M2 area). (**C**) Numbers of DEGs at 1 week, 4 weeks and 8 weeks post-stroke. (**D**) Volume plot of transcriptome gene expression in penumbra at 1 week post-stroke. (*x*-axis: Volume, *y*-axis: log2 fold change). Red dots are top five ranking upregulated genes, fold change >2 and highest volume. Blue dots for top five ranking downregulated genes, fold change <−2 and highest volume. (**E**) Volume plot for 4 weeks post-stroke. (**F**) Volume plot for 8 weeks post-stroke.

**Figure 3 ijms-20-06349-f003:**
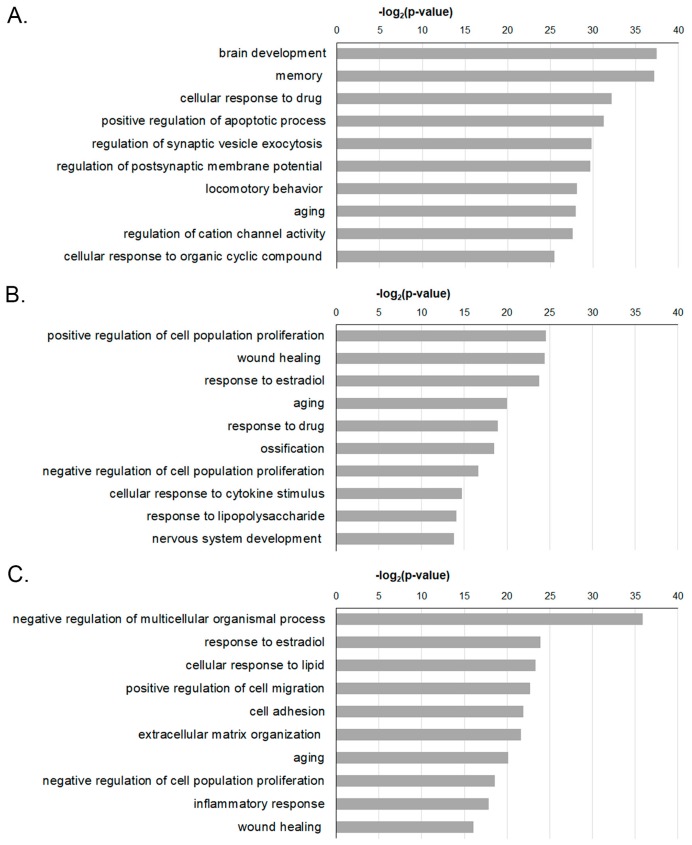
Gene ontology enrichment analysis of DEGs in the penumbra area. Biological processes in gene ontology analysis with most significant *p*-value were shown for (**A**) 1 week, (**B**) 4 weeks, and (**C**) 8 weeks post-stroke.

**Figure 4 ijms-20-06349-f004:**
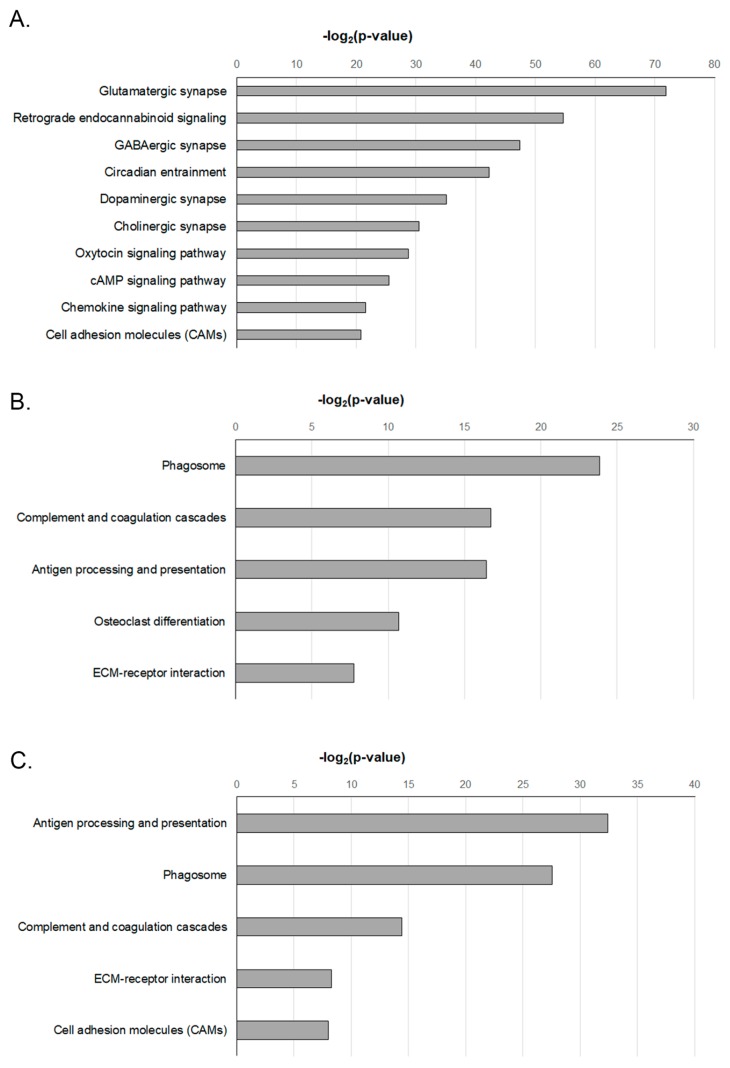
KEGG pathway analysis of DEGs with most significant *p*-value for (**A**) 1 week, (**B**) 4 weeks, and (**C**) 8 weeks post-stroke in penumbra.

**Figure 5 ijms-20-06349-f005:**
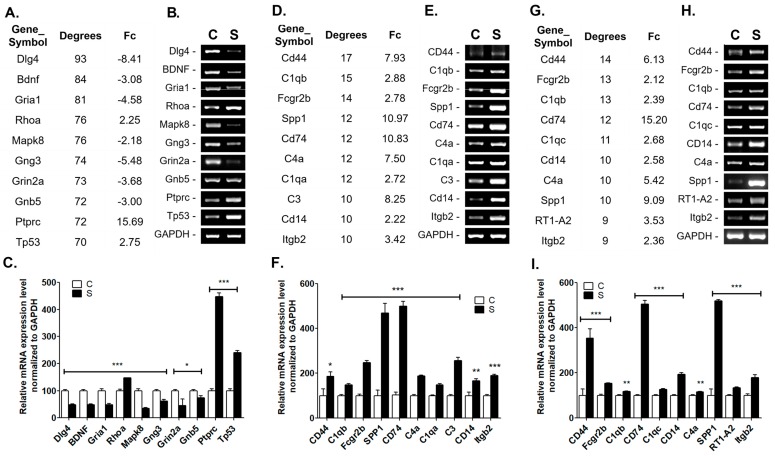
Top 10 genes with higher degrees of PPI network analysis using StringTie. Validation was achieved using reverse-transcription polymerase chain reaction (RT-PCR). The top 10 genes with higher degrees of PPI network analysis and fold change (Fc) of RNA-sequencing analysis, validation of mRNA expression level using RT-PCR, and comparison of relative mRNA expression level from the penumbra between control at 1 week (**A**–**C**), 4 weeks (**D**–**F**) and 8 weeks (**G**–**I**) post-stroke, respectively. *n* = 6 per group, * *p* < 0.05, ** *p* < 0.01, *** *p* < 0.001 compared to control (**C**); Stroke (**S**).

**Table 1 ijms-20-06349-t001:** Top-five ranking of genes list.

1 week post stroke					
**Gene_ID**	**Gene_Symbol of Rats**	**Description**	**Fold Change**	**log2(Fc)**	**Volume**
50665	*Tmsb10*	thymosin, beta 10	2.98	1.57	10.03
29292	*Ftl1*	ferritin light chain 1	7.36	2.88	9.96
64529	*Ctsb*	cathepsin B	7.31	2.87	9.46
171293	*Ctsd*	cathepsin D	12.36	3.63	9.29
24223	*B2m*	beta-2 microglobulin	3.28	1.71	8.77
24547	*Mbp*	myelin basic protein	−22.65	−3.32	10.88
25742	*S100b*	S100 calcium binding protein B	−6.88	−2.32	9.76
287005	*Camk2n1*	calcium/calmodulin-dependent protein kinase II inhibitor 1	−48.01	−4.04	9.39
24191	*Aldoc*	aldolase, fructose-bisphosphate C	−9.07	−2.80	9.03
171114	*Ndrg2*	NDRG family member 2	−7.55	−2.50	8.72
4 week post stroke					
**Gene_ID**	**Gene_Symbol of Rats**	**Description**	**Fold Change**	**log2(Fc)**	**Volume**
25526	*Ptgds*	prostaglandin D2 synthase	13.45	1.58	9.77
24387	*Gfap*	glial fibrillary acidic protein	8.65	1.73	8.36
80841	*Fabp7*	fatty acid binding protein 7	4.86	2.88	7.00
286898	*Npc2*	NPC intracellular cholesterol transporter 2	2.84	2.92	6.86
94270	*Nnat*	neuronatin	4.69	2.90	6.71
362220	*Lamp5*	lysosomal-associated membrane protein family, member 5	−2.88	−1.77	6.73
24587	*Nefh*	neurofilament, heavy polypeptide	−3.72	−1.41	5.54
315611	*Scn4b*	sodium voltage-gated channel beta subunit 4	−3.82	−1.85	4.57
108348108	*LOC108348108*	heat shock 70 kDa protein 1A	−5.71	−2.57	3.89
363168	*Lyzl4*	lysozyme-like 4	−3.89	−1.60	3.42
8 week post stroke					
**Gene_ID**	**Gene_Symbol of Rats**	**Description**	**Fold Change**	**log2(Fc)**	**Volume**
25526	*Ptgds*	prostaglandin D2 synthase	10.31	3.33	10.35
24387	*Gfap*	glial fibrillary acidic protein	6.08	2.44	8.86
81818	*Vim*	vimentin	10.17	3.13	6.96
25211	*Lyz2*	lysozyme 2	5.49	2.35	6.32
56646	*Lgals1*	galectin 1	4.47	2.05	5.98
24587	*Nefh*	neurofilament, heavy polypeptide	−3.17	−1.64	5.41
314627	*Plk5*	polo-like kinase 5	−2.60	−1.37	3.46
315564	*Robo3*	roundabout guidance receptor 3	−3.49	−1.72	3.42
117023	*Kcns1*	potassium voltage-gated channel, modifier subfamily S, member 1	−3.57	−1.81	2.99
24805	*Syt2*	synaptotagmin 2	−5.61	−2.39	2.13

**Table 2 ijms-20-06349-t002:** The top 10 genes in the protein–protein interaction (PPI) network of biological processes in gene ontology analysis.

Time	1 Week		4 Weeks		8 Weeks		Time	1 Week		4 Weeks		8 Weeks	
**GO**	**Brain Development (234 Genes)**		**Positive Regulation of Cell Population Proliferation (57 Genes)**		**Negative Regulation of Multicellular Organismal Process (73 Genes)**		**GO**	**Regulation of Postsynaptic Membrane Potential (58 Genes)**		**Ossification (24 Genes)**		**Extracellular Matrix Organization (25 Genes)**	
	**Gene**	**Fc**	**Gene**	**Fc**	**Gene**	**Fc**		**Gene**	**Fc**	**Gene**	**Fc**	**Gene**	**Fc**
1	*Tp53*	2.75	*Igf1*	2.35	*Cd44*	6.13	1	*Gria1*	−4.58	*Igf1*	2.35	*Col1a1*	13.87
2	*Casp3*	2.40	*Ptprc*	2.64	*Timp1*	6.30	2	*Grin2a*	−3.68	*Spp1*	10.97	*Col3a1*	13.73
3	*Rhoa*	2.25	*Timp1*	7.38	*Ccl2*	2.66	3	*Grm5*	−3.18	*Mmp2*	2.17	*Lum*	2.92
4	*Gnai1*	-2.84	*Cxcl12*	2.13	*Tgfb1*	2.05	4	*Gabra1*	−26.42	*Col1a1*	15.05	*Postn*	−2.40
5	*Syt1*	-20.35	*Mmp2*	2.17	*Spp1*	9.09	5	*Gabrg2*	−12.33	*Sparc*	2.12	*Fbln1*	6.40
6	*Grin2a*	-3.68	*Tgfb1*	2.09	*Col3a1*	13.73	6	*Grin2b*	−2.39	*Igf2*	25.06	*Mmp14*	6.88
7	*Ccr5*	3.14	*Lgals3*	36.53	*Vim*	10.17	7	*Gabrb2*	−6.06	*Tgfb1*	2.09	*Fmod*	7.47
8	*Myc*	5.49	*Rac2*	2.59	*Dcn*	6.11	8	*Gria2*	−8.97	*Igfbp5*	2.65	*Tgfb1*	2.05
9	*Aif1*	10.36	*Hmox1*	2.39	*Cd74*	15.20	9	*Grm1*	−2.47	*Ctsk*	3.21	*Lgals3*	21.69
10	*Gnao1*	-5.73	*Igf2*	25.06	*Anxa1*	10.74	10	*Grik1*	−3.15	*Twist1*	2.44	*Col18a1*	3.03
**GO**	**Memory (76 Genes)**		**Wound Healing (32 Genes)**		**Response to Estradiol (27 Genes)**		**GO**	**Locomotory Behavior (91 Genes)**		**Negative Regulation of Cell Population Proliferation (41 genes)**		**Aging (34 Genes)**	
	**Gene**	**Fc**	**Gene**	**Fc**	**Gene**	**Fc**		**Gene**	**Fc**	**Gene**	**Fc**	**Gene**	**Fc**
1	*Bdnf*	−3.08	*Timp1*	7.38	*Ccl2*	2.66	1	*Gad1*	−3.54	*Igf1*	2.35	*Ccl2*	2.66
2	*Gria1*	−4.58	*Igf1*	2.35	*Col1a1*	13.87	2	*Scn1a*	−9.28	*Cd44*	7.93	*Tgfb1*	2.05
3	*Grin2b*	−2.39	*Sparc*	2.12	*Igf2*	28.16	3	*Snap25*	−226.77	*Tgfb1*	2.09	*Timp1*	6.30
4	*Grin2a*	−3.68	*Tgfb1*	2.09	*Tgfb1*	2.05	4	*Ppp1r1b*	−4.62	*Sparc*	2.12	*Icam1*	2.10
5	*Slc17a7*	−134.52	*Col1a1*	15.05	*Vim*	10.17	5	*Adora2a*	−2.65	*Tgfbr2*	2.58	*Clu*	2.29
6	*Cnr1*	−3.65	*Dcn*	4.81	*Nqo1*	2.16	6	*Mapk8*	−2.18	*Lgals3*	36.53	*Gpx1*	2.09
7	*Egr1*	−3.21	*Cd44*	7.93	*Anxa1*	10.74	7	*Gnao1*	−5.73	*Hmox1*	2.39	*Nqo1*	2.16
8	*Htr2a*	−3.24	*Igf2*	25.06	*Igfbp3*	5.34	8	*Cacna1a*	−5.10	*Bmp7*	5.42	*Itgb2*	2.36
9	*Snap25*	−226.77	*Col3a1*	18.13	*Cd4*	2.38	9	*Grm5*	−3.18	*Timp2*	2.29	*Serping1*	16.63
10	*Cck*	−98.24	*Pf4*	2.86	*Pdgfra*	2.33	10	*Grm1*	−2.47	*Gpc3*	4.84	*Mbp*	−3.01
**GO**	**Cellular Response to Drug (140 Genes)**		**Response to Estradiol (27 Genes)**		**Cellular Response to Lipid (43 Genes)**		**GO**	**Aging (138 Genes)**		**Cellular Response to Cytokine Stimulus (42 Genes)**		**Negative Regulation of Cell Population Proliferation (42 Genes)**	
	**Gene**	**Fc**	**Gene**	**Fc**	**Gene**	**Fc**		**Gene**	**Fc**	**Gene**	**Fc**	**Gene**	**Fc**
1	*Tp53*	2.75	*Igf1*	2.35	*Ccl2*	2.66	1	*Tp53*	2.75	*Cxcl12*	2.13	*Cd44*	6.13
2	*Mapk8*	−2.18	*Col1a1*	15.05	*Icam1*	2.10	2	*Igf1*	52.19	*Cd44*	7.93	*Lgals3*	21.69
3	*Ccl2*	22.32	*Igf2*	25.06	*Cd40*	2.28	3	*Stat3*	2.55	*Mmp2*	2.17	*Ccl2*	2.66
4	*Myc*	5.49	*Mmp2*	2.17	*Cxcl13*	2.31	4	*Agt*	−18.17	*Spp1*	10.97	*Tgfb1*	2.05
5	*Icam1*	6.54	*Tgfb1*	2.09	*Tgfb1*	2.05	5	*Mmp9*	17.34	*Col1a1*	15.05	*Tgfbr2*	2.19
6	*Igf1*	52.19	*Vim*	13.22	*Il18*	2.36	6	*Bdnf*	−3.08	*Ifitm1*	34.44	*Aif1*	2.16
7	*Hmox1*	52.12	*Bmp7*	5.42	*Spp1*	9.09	7	*Icam1*	6.54	*Cd74*	10.83	*Fcgr2b*	2.12
8	*Rhoa*	2.25	*Igfbp3*	2.12	*Col1a1*	13.87	8	*Ccl2*	22.32	*Ifitm3*	4.49	*Gpc3*	3.97
9	*Il18*	10.22	*Pdgfra*	2.66	*Cxcl16*	6.41	9	*Fos*	−2.45	*Gbp2*	4.47	*Tspo*	2.79
10	*Gnai2*	2.31	*Igfbp2*	14.03	*Pf4*	3.40	10	*Hras*	−2.11	*Cyba*	4.20	*Inhba*	−2.18
**GO**	**Positive Regulation of Apoptotic Process (181 Genes)**		**Aging (34 Genes)**		**Positive Regulation of Cell Migration (39 Genes)**		**GO**	**Regulation of Cation Channel Activity (73 Genes)**		**Response to Lipopopysaccharide (30 Genes)**		**Inflammatory Response (31 Genes)**	
	**Gene**	**Fc**	**Gene**	**Fc**	**Gene**	**Fc**		**Gene**	**Fc**	**Gene**	**Fc**	**Gene**	**Fc**
1	*Tp53*	2.75	*Igf1*	2.35	*Cxcl12*	2.44	1	*Dlg4*	−8.41	*Igf1*	2.35	*Ccl2*	2.66
2	*Casp3*	2.40	*Tgfb1*	2.09	*Ccl2*	2.66	2	*Cacna1a*	−5.10	*Spp1*	10.97	*Icam1*	2.10
3	*Mapk8*	−2.18	*Timp1*	7.38	*Icam1*	2.10	3	*Cacng8*	−9.08	*Col1a1*	15.05	*Timp1*	6.30
4	*Myc*	5.49	*Mmp2*	2.17	*Cxcl13*	2.31	4	*Cacng7*	−4.68	*Fos*	−2.66	*Cd44*	6.13
5	*Tlr4*	3.46	*Fos*	-2.66	*Ccl19*	3.69	5	*Shank1*	−28.89	*Tgfbr2*	2.58	*Spp1*	9.09
6	*Ccl2*	22.32	*Clu*	2.19	*Col1a1*	13.87	6	*Rasgrf1*	−14.10	*Sparc*	2.12	*Ccl19*	3.69
7	*Mmp9*	17.34	*Dcn*	4.81	*Anxa1*	10.74	7	*Cacnb1*	−6.68	*Tgfb1*	2.09	*Pf4*	3.40
8	*Casp8*	3.39	*Serping1*	21.82	*Cd40*	2.28	8	*Cacnb3*	−4.16	*Bmp7*	5.42	*Itgb2*	2.36
9	*Agt*	−18.17	*Vim*	13.22	*Il18*	2.36	9	*Cacnb2*	−2.62	*C3*	8.25	*Il18*	2.36
10	*Anxa5*	2.05	*Ucp2*	2.75	*Aif1*	2.16	10	*Cacnb4*	−4.46	*Cyba*	4.20	*Cxcl13*	2.31
**GO**	**Regulation of Synaptic Vesicle Exocytosis (59 Genes)**		**Response to Drug (64 Genes)**		**Cell Adhesion (44 genes)**		**GO**	**Cellular Response to Organic Cyclic Compound (166 Genes)**		**Nervous System Development (90 Genes)**		**Wound Healing (28 Genes)**	
	**Gene**	**Fc**	**Gene**	**Fc**	**Gene**	**Fc**		**Gene**	**Fc**	**Gene**	**Fc**	**Gene**	**Fc**
1	*Syt1*	−20.35	*Mmp2*	2.17	*Cd44*	6.13	1	*Casp3*	2.40	*Cd44*	7.93	*Timp1*	6.30
2	*Stxbp1*	−7.15	*Igf1*	2.35	*Spp1*	9.09	2	*Stat3*	2.55	*Igf1*	2.35	*Col1a1*	13.87
3	*Stx1a*	−23.65	*Col1a1*	15.05	*Cxcl12*	2.44	3	*Igf1*	52.19	*Mmp2*	2.17	*Tgfb1*	2.05
4	*Rab3a*	−23.39	*Tgfb1*	2.09	*Icam1*	2.10	4	*Myc*	5.49	*Spp1*	10.97	*Dcn*	6.11
5	*Cplx1*	−62.03	*Fos*	-2.66	*Col3a1*	13.73	5	*Bdnf*	-3.08	*Cxcl12*	2.13	*Cd44*	6.13
6	*Syt2*	−8.04	*Hmox1*	2.39	*Lamb2*	2.15	6	*Ccl2*	22.32	*Gfap*	8.65	*Col3a1*	13.73
7	*Vamp1*	−14.02	*Igf2*	25.06	*Itgb4*	10.95	7	*Rhoa*	2.25	*Col3a1*	18.13	*Igf2*	28.16
8	*Cplx2*	−18.49	*Sparc*	2.12	*Thbs2*	3.49	8	*Egr1*	−3.21	*Fos*	−2.66	*Postn*	−2.40
9	*Syp*	−44.39	*Vim*	13.22	*Tgfbr2*	2.19	9	*Icam1*	6.54	*Vim*	13.22	*Vim*	10.17
10	*Camk2a*	−139.81	*Gpx1*	2.17	*Postn*	-2.40	10	*Actb*	2.25	*Mmp14*	6.52	*Pf4*	3.40

**Table 3 ijms-20-06349-t003:** DEGs of Top 10 degrees in PPI analysis.

1 week post stroke			
**Gene_Symbol**	**Description**	**Degrees**	**Fc**
*Dlg4*	discs large MAGUK scaffold protein 4	93	−8.41
*Bdnf*	brain-derived neurotrophic factor	84	−3.08
*Gria1*	glutamate ionotropic receptor AMPA type subunit 1	81	−4.58
*Rhoa*	ras homolog family member A	76	2.25
*Mapk8*	mitogen-activated protein kinase 8	76	−2.18
*Gng3*	G protein subunit gamma 3	74	−5.48
*Grin2a*	glutamate ionotropic receptor NMDA type subunit 2A	73	−3.68
*Gnb5*	G protein subunit beta 5	72	−3
*Ptprc*	protein tyrosine phosphatase, receptor type, C	72	15.69
*Tp53*	tumor protein p53	70	2.75
4 weeks post stroke			
**Gene_Symbol**	**Description**	**Degrees**	**Fc**
*Cd44*	CD44 molecule (Indian blood group)	17	7.93
*C1qb*	complement component 1, q subcomponent, B chain	15	2.88
*Fcgr2b*	Fc fragment of IgG, low affinity IIb, receptor	14	2.78
*Spp1*	secreted phosphoprotein 1	12	10.97
*Cd74*	CD74 molecule	12	10.83
*C4a*	complement component 4A (Rodgers blood group)	12	7.5
*C1qa*	complement component 1, q subcomponent, A chain	12	2.72
*C3*	complement component 3	10	8.25
*Cd14*	CD14 molecule	10	2.22
*Itgb2*	integrin subunit beta 2	10	3.42
8 weeks post stroke			
**Gene_Symbol**	**Description**	**Degrees**	**Fc**
*Cd44*	CD44 molecule (Indian blood group)	14	6.13
*Fcgr2b*	Fc fragment of IgG, low affinity IIb, receptor	13	2.12
*C1qb*	complement component 1, q subcomponent, B chain	13	2.39
*Cd74*	CD74 molecule	12	15.2
*C1qc*	complement component 1, q subcomponent, C chain	11	2.68
*Cd14*	CD14 molecule	10	2.58
*C4a*	complement component 4A (Rodgers blood group)	10	5.42
*Spp1*	secreted phosphoprotein 1	10	9.09
*RT1-A2*	RT1 class Ia, locus A2	9	3.53
*Itgb2*	integrin subunit beta 2	9	2.36

**Table 4 ijms-20-06349-t004:** Primers used in RT-PCR.

Gene Symbol of Rats	Forward Primer	Reverse Primer
*Dlg4*	ATGCCTACCTGAGTGACAGC	CCCAGCAAGGATGAAGGAGA
*Bdnf*	AGGTTCGAGAGGTCTGACGA	GCTGTGACCCACTCGCTAAT
*Gria1*	AAGCACGTGGGCTACTCCTA	GACGACGCTCACTCCAATGT
*Rhoa*	TCCATCGACAGCCCTGATAG	CTTTTCTTCCCGCGTCTAGC
*Mapk8*	CTACAACCAACAGTAAGGAC	GTTTCCACTCCTCTATTGTG
*Gng3*	GACCCCCGTTAACAGCACTA	GAAGTGGGCACAGGAGTGAT
*Grin2a*	TGGTGATGGTGAGATGGAGG	GTGTACCCCATGGATGCAAC
*Gnb5*	GTCTGTCGCTATGCACACCA	AGCATCTCCACTGGGGTAGT
*Ptprc*	GATTGCCGATGAGGGTAGAC	CATCAACTGTCTCATCCCGG
*Tp53*	GCCCATCCTTACCATCATCA	GCACGGGCATCCTTTAATTC
*CD44*	GACAGAAACAGCACCAGTGC	CTTGGATGGTTGTTGTGGGC
*C1qb*	TGATGGCAAACCAGGCACTC	CCTTTTCGAAGCGAATGGCC
*Fcgr2b*	TGGGAGTGATTTCTGACTGG	GCTACAATCGTCAATACCGG
*Spp1*	CAGGAGTCCGATGAGGCTAT	CCTCATGGCTGTGAAACTCG
*Cd74*	CAGGCCACCACTGCTTACTT	TGTGCTTCAGATTCTCCGGG
*C4a*	GCCCAGCAAGTATCAGTGCC	CAGTCAGGGTAGGGGCCAAA
*C1qa*	CTCAGCTATTCGGCAGAACC	CCTTCTCAATCCACACCTCG
*C3*	CTTCATGGACTGCTGCAACT	TCTGCCACACAGATCCCTTT
*Cd14*	TTTCTTGCAAACAGGTCGGC	AGCAGTATCCCGCAGTGAAT
*Itgb2*	ACACCCATCCCGAGAAGCTG	CCATCGTTGGGGGTCAGGAT
*C1qc*	GATGGGCATGATGGACTTCA	GTGTGTTGTAATCCCCCTGA
*RT1-A2*	CCAGGACATGGAGCTTGTGG	CTGGAGCAGGGGTGTAGTCA
*GAPDH*	ATCACCATCTTCCAGGAGCG	GATGGCATGGACTGTGGTCA

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
