# Peer review of "Sequential Transcriptome Changes in the Penumbra after Ischemic Stroke"

_ijms, 2019, doi:10.3390/ijms20246349_

Round 1

Reviewer 1 Report

The present manuscript is well written. The study is designed and performed properly. The methodology and results are written in detail. Authors discussed the findings from present study properly. Still, there is need to address following queries

Why authors have not used female rats in the present study?  Did authors perform power analysis to determine sample size in the present study? If yes, explain at appropriate place in methodology section.   Did authors perform the study blindly? In Fig 1C, authors have not included mNSS score for normal rats for the comparison. This needs explanation.  

Author Response

We are grateful to reviewer’s insightful and instructive comments and have made significant revisions accordingly. The most of points raised by reviewer were indeed valid and thus might be constructive to further strengthen our manuscript. Below is our point-by-point response to reviewer’s comments.

Comments and Suggestions for Authors

The present manuscript is well written. The study is designed and performed properly. The methodology and results are written in detail. Authors discussed the findings from present study properly. Still, there is need to address following queries.

1.Why authors have not used female rats in the present study?  

Epidemiological data indicate that stroke rates are strikingly sex-specific, exhibiting incidence rates that are high in men compared with women, regardless of ethnicity [1]. Several animal studies have reported the role of female sex steroids in protecting ischemic brain injury [2-4]. Therefore, male rats were used instead of female rat because of the high incidence of stroke in men and the protective effects of female hormones. However, since there are gender differences in the pathogenesis and treatment mechanism of the stroke, it would be better to conduct experiments on both sexes in further studies.

The following sentences are included in the materials and methods section.

They also followed the criteria for Stroke Therapy Academic Industry Roundtable (STAIR) for preclinical stroke investigations. Adult male wistar rats, 8 weeks old (283.88 ± 2.06 g), were purchased from the Orient Bio Incorporation (Seongnam, Korea). All rats were housed in a temperature-controlled room (23 ± 0.5℃) with 12h:12h light/dark cycle. Food and water add libitum. Male rats were used because of the high incidence of stroke in men and to rule out the protective effects of female hormones [1-4].

Appelros, P.; Stegmayr, B.; Terent, A. Sex differences in stroke epidemiology: a systematic review. Stroke 2009, 40, 1082-1090, doi:10.1161/STROKEAHA.108.540781. Herson, P.S.; Traystman, R.J. Animal models of stroke: translational potential at present and in 2050. Future Neurol 2014, 9, 541-551, doi:10.2217/fnl.14.44. Toung, T.K.; Hurn, P.D.; Traystman, R.J.; Sieber, F.E. Estrogen decreases infarct size after temporary focal ischemia in a genetic model of type 1 diabetes mellitus. Stroke 2000, 31, 2701-2706, doi:10.1161/01.str.31.11.2701. Vannucci, S.J.; Willing, L.B.; Goto, S.; Alkayed, N.J.; Brucklacher, R.M.; Wood, T.L.; Towfighi, J.; Hurn, P.D.; Simpson, I.A. Experimental stroke in the female diabetic, db/db, mouse. J Cereb Blood Flow Metab 2001, 21, 52-60, doi:10.1097/00004647-200101000-00007.

2.Did authors perform power analysis to determine sample size in the present study?

If yes, explain at appropriate place in methodology section.  

The following sentences are included in the materials and methods section.

Based on previous infarct volume data using an identical surgical design (photothrombotic ischemic surgery), a sample size calculation (power = 0.8; alpha = 0.05) estimated 6 animals per group using G-power version 3.1.9.4. For RNA sequencing, 5 animals were randomly selected to each groups and 6 animals/group were using for infarction measurement and modified neurological severity score (mNSS) (Figure 1A).

3.Did authors perform the study blindly?

Animals and samples were number-coded and investigators were always blinded to the experimental groups until the end of the data analysis.

The above sentence is included in the materials and methods section.

4.In Fig 1C, authors have not included mNSS score for normal rats for the comparison. This needs explanation.  

Thank you for your comments. It was written incorrectly. There are corrected.

The following sentences are included in the results section.

mNSS scores ranged from 0 up to 14. High scores indicate that the rats had more neurological deficits from stroke. At 1, 4, and 8 weeks after stroke, the mNSS scores were 11.0 + 0.0, 10.7 + 0.52, and 11.0 + 0.0, respectively, with no difference between 1, 4, and 8 weeks after stroke (Figure 1C).

Reviewer 2 Report

Figure 1c should be corrected as the changes indicated in the text „the score on the mNSS (…) was increased after stroke (…)” were not shown (lack of control bar) and statistical significance indicators.

In the results part the authors wrote that examined groups included “(1) 1-, 4-, and 8-week control animals and contralateral cortex 4 and 8 weeks after stroke; (…)” meanwhile later the authors state that “At 1, 4, and 8 weeks after stroke, the contralateral cortex exhibited (…)”. The text should be corrected as according to the text contralateral cortex at week 4 and 8 after stroke was analyzed not at week 1.

The authors should add references after this statement “Although several studies have shown that ischemic stroke may affect the profiling of gene expression in the brain, (…)”

The text in the part “4.3. Determination of the Infarction Volume” should be formatted to unify its presentation.

Author Response

We are grateful to reviewer’s insightful and instructive comments and have made significant revisions accordingly. The most of points raised by reviewer were indeed valid and thus might be constructive to further strengthen our manuscript. Below is our point-by-point response to reviewer’s comments.

1.Figure 1c should be corrected as the changes indicated in the text „the score on the mNSS (…) was increased after stroke (…)” were not shown (lack of control bar) and statistical significance indicators.

Thank you for your comments. It was written incorrectly. There are corrected.

The following sentences are included in the results section.

mNSS scores ranged from 0 up to 14. High scores indicate that the rats had more neurological deficits from stroke. At 1, 4, and 8 weeks after stroke, the mNSS scores were 11.0 + 0.0, 10.7 + 0.52, and 11.0 + 0.0, respectively, with no difference between 1, 4, and 8 weeks after stroke (Figure 1C).

2.In the results part the authors wrote that examined groups included “(1) 1-, 4-, and 8-week control animals and contralateral cortex 4 and 8 weeks after stroke; (…)” meanwhile later the authors state that “At 1, 4, and 8 weeks after stroke, the contralateral cortex exhibited (…)”. The text should be corrected as according to the text contralateral cortex at week 4 and 8 after stroke was analyzed not at week 1.

Thank you for your comments. It was written incorrectly. There are corrected.

The following sentences are included in the results section.

According to heatmap analysis, the experimental groups were as follows: (1) 1-, 4-, and 8-week control cortex animals and contralateral cortex 1, 4 and 8 weeks after stroke; (2) ipsilateral cortex 4 and 8 weeks after stroke; and (3) ipsilateral cortex 1 week after stroke. Gene expression patterns in the contralateral cortex of 1, 4, 8 weeks after stroke exhibited were similar to the control group. Therefore, the contralateral cortices were excluded from subsequent analyses, and the changes in mRNA expression on the ipsilateral injured side were mainly analyzed.

3.The authors should add references after this statement “Although several studies have shown that ischemic stroke may affect the profiling of gene expression in the brain, (…)”

Thank you for your advice. There are added cited references.

Although several studies have shown that ischemic stroke may affect the profiling of gene expression in the brain [18-22],

Zhang, C.; Zhu, Y.; Wang, S.; Zachory Wei, Z.; Jiang, M.Q.; Zhang, Y.; Pan, Y.; Tao, S.; Li, J.; Wei, L. Temporal Gene Expression Profiles after Focal Cerebral Ischemia in Mice. Aging and disease 2018, 9, 249-261, doi:10.14336/AD.2017.0424 [doi]. Buga, A.M.; Margaritescu, C.; Scholz, C.J.; Radu, E.; Zelenak, C.; Popa-Wagner, A. Transcriptomics of post-stroke angiogenesis in the aged brain. Frontiers in aging neuroscience 2014, 6, 44. Li, S.; Overman, J.J.; Katsman, D.; Kozlov, S.V.; Donnelly, C.J.; Twiss, J.L.; Giger, R.J.; Coppola, G.; Geschwind, D.H.; Carmichael, S.T. An age-related sprouting transcriptome provides molecular control of axonal sprouting after stroke. Nature neuroscience 2010, 13, 1496. Buga, A.M.; Sascau, M.; Pisoschi, C.; Herndon, J.G.; Kessler, C.; Popa-Wagner, A. The genomic response of the ipsilateral and contralateral cortex to stroke in aged rats. Journal of Cellular and Molecular Medicine 2008, 12, 2731-2753, doi:10.1111/j.1582-4934.2008.00252.x [doi]. Ito, M.; Aswendt, M.; Lee, A.G.; Ishizaka, S.; Cao, Z.; Wang, E.H.; Levy, S.L.; Smerin, D.L.; McNab, J.A.; Zeineh, M., et al. RNA-Sequencing Analysis Revealed a Distinct Motor Cortex Transcriptome in Spontaneously Recovered Mice After Stroke. Stroke 2018, 49, 2191-2199, doi:10.1161/STROKEAHA.118.021508.

4.The text in the part “4.3. Determination of the Infarction Volume” should be formatted to unify its presentation.

The intact areas of ipsilateral and contralateral hemispheres were measured using image J then the volume of intact hemisphere was calculated (intact area*0.04*20) and summed among slices. Total infarction volume was determined: the volume of contralateral hemisphere – the volume of intact area in ipsilateral hemisphere [23,35].